# Integrated Analysis of Somatic DNA Variants and DNA Methylation of Tumor Suppressor Genes in Colorectal Cancer

**DOI:** 10.3390/ijms26041642

**Published:** 2025-02-14

**Authors:** Hisashi Nishiki, Hiroki Ura, Sumihito Togi, Hisayo Hatanaka, Hideto Fujita, Hiroyuki Takamura, Yo Niida

**Affiliations:** 1General and Digestive Surgery, Kanazawa Medical University, Uchinada 920-0293, Japan; hisashi@kanazawa-med.ac.jp (H.N.); hfujita@kanazawa-med.ac.jp (H.F.); takamuh@kanazawa-med.ac.jp (H.T.); 2Center for Clinical Genomics, Kanazawa Medical University Hospital, Uchinada 920-0293, Japantogi@kanazawa-med.ac.jp (S.T.);; 3Division of Genomic Medicine, Department of Advanced Medicine, Medical Research Institute, Kanazawa Medical University, Uchinada 920-0293, Japan

**Keywords:** colorectal cancer, tumor suppressor gene, DNA methylation, somatic variant, loss of heterozygosity (LOH), next-generation sequencing (NGS)

## Abstract

DNA methylation of tumor suppressor genes in cancer is known to be a mechanism for silencing gene expression, but much remains unknown about its extent and relationship to somatic variants at the DNA sequence level. In this study, we comprehensively analyzed DNA methylation and somatic variants of all gene regions across the genome of the major tumor suppressor genes, *APC*, *TP53*, *SMAD4*, and mismatch repair genes in colorectal cancer using a novel next-generation sequencing-based analysis method. The Targeted Methyl Landscape (TML) shows that DNA hypermethylation patterns of these tumor suppressor genes in colorectal cancer are more complex and widespread than previously thought. Extremely high levels of DNA methylation were observed in relatively long regions around exon 1A of *APC* and exon 1 and surrounding region of *MLH1*. DNA hypermethylation occurred whether or not somatic DNA variants were present in the tumor. Even in tumors where the loss of heterozygosity has been demonstrated by somatic variants alone, additional methylation of the same gene can occur. Our data demonstrate that somatic variants and hypermethylation of these tumor suppressor genes were considered independent, parallel events, not exclusive of each other or having one event affecting the other.

## 1. Introduction

Since Knudson’s two-hit hypothesis, it has been assumed that loss of function of both alleles of a tumor suppressor gene promotes carcinogenesis, not only in hereditary but also in sporadic cancers [1,2]. The mechanism of loss of tumor suppressor gene function is thought to be primarily due to variants in the DNA sequence, and technological innovations have made it possible to study this in detail in clinical practice [3]. On the other hand, epigenetic modifications that do not involve changes in DNA sequences are also known for gene silencing mechanisms, and, among them, DNA methylation is one of the best known mechanisms for tumor suppressor gene inactivation in cancer [4,5,6]. Whether the underlying cause is DNA variants or methylation, either in combination or alone, if a specific tumor suppressor gene is inactivated in both alleles, there is a possibility that cancer will be promoted. However, it is not clear how these two mechanisms interact with each other to synergistically promote cancer, or whether they act completely independently of each other. In addition, the current content of DNA methylation analysis is not sufficient. Research on DNA methylation has typically focused on the gene promoter located in the CpG island upstream of exon 1 [7,8,9,10], but it is unclear which regions of DNA are actually altered in methylation status in cancer. There are studies that use next-generation sequencing (NGS) to perform genome wide DNA methylation analysis of cancer to find prognostic markers [11,12], and there are studies that attempt to obtain an overview of DNA variants and methylation across the whole genome [13], but there are no studies that focus on specific tumor suppressor genes and perform in-depth analyses. To date, no study has simultaneously analyzed somatic DNA variants and methylation in the entire genomic region where a specific tumor suppressor gene is located and elucidated the relationship between them. As a result, the true nature of the complex state of these two elements in specific genes remains unclear.

In colorectal cancer, a multi-step genetic variant-based mechanism of carcinogenesis has been demonstrated and the major tumor suppressor genes involved have been identified [14,15,16]. All colorectal cancers begin as benign adenomas and are accelerated by successive somatic gene variants that promote malignant progression. Activation of the WNT signaling pathway is essential for the initiation of colorectal cancer, and the *APC* (APC regulator of WNT signaling pathway) gene on chromosome 5q22.2 acts as a gatekeeper for this pathway. Biallelic inactivation of the *APC* gene, loss of heterozygosity (LOH), is required for the development of colorectal cancer, and germline mutations in *APC* cause familial adenomatous polyposis [15]. DNA methylation may be a second hit mutation in *APC* in sporadic colorectal cancer, but details are unknown. Hypermethylation of other tumor suppressor genes has also been reported in colorectal cancer [17,18,19]. Gain-of-function variants of *KRAS* and inactivation of *TP53* are frequently involved in chromosomal instability, CIN (84%). On the other hand, inactivation of mismatch repair (MMR) genes (*MSH2*, *MSH6*, *MLH1*, and *PMS2*) is associated with microsatellite instability high (MSI-H) and hypermutation (13%), and germline variants in MMR genes cause Lynch syndrome (hereditary nonpolyposis colon cancer) [20]. MSI-H or mismatch repair-deficient (dMMR) colorectal cancer is characterized by high expression of PD-L1 (programmed death ligand-1) and is therefore amenable to treatment with anti-PD-1 (programmed cell death protein-1)/PD-L1 checkpoint inhibitors, a type of immunomodulatory therapy [21]. Immunohistochemistry for dMMR or MSI testing of DNA is recommended for evaluation of MMR deficiency but can be difficult to interpret in some cases. It has been reported that dMMR results may not be consistent with germline mutations, and it has been suggested that the cause may depend on the status of somatic mutations [22]. We thought that colorectal cancer was a suitable subject for DNA methylation analysis as a second hit because the molecular mechanisms leading to carcinogenesis are relatively clear and it is also clinically important to understand the mechanisms of the second hit.

In this study, we develop a novel approach using NGS to simultaneously analyze somatic variants and DNA methylation in key colorectal cancer tumor suppressor genes across the entire genomic region where the genes reside. The objectives of this study are to verify the utility of these new NGS techniques and to clarify the nature of somatic variants that inactivate tumor suppressor genes, the extent and degree of cancer-specific DNA methylation, and the relationships between them. To achieve this, we performed a comparative analysis of DNA extracted from colorectal cancer tissue and normal colon tissue obtained from surgical specimens from 32 patients for *APC*, *TP53*, *SMAD4*, *MSH2*, *MSH6*, *MLH1*, and *PMS2* genes using these integrated methods.

## 2. Results

### 2.1. Clinical Profile of the Patients

The clinical profile of the 32 patients with colorectal cancer in this study is shown in Table 1. The mean age at the time of colectomy was 72.7 years, and the male to female ratio was 1:1. Postoperative pathologic examination revealed *KRAS* variants in 12 patients, *BRAF* variants in 4 patients, and MSI-H in 5 patients. The location of the tumors was variable, and no consistent trends could be observed.

### 2.2. Overview of Analysis Results for Each Library

Very long amplicon sequencing (vLAS), targeted genome capture (TgCAp) and targeted methyl landscape (TML) all achieved continuous coverage of the target area. Continuity of coverage was better with vLAS than with TgCap and TML. In vLAS, equimolar mixtures of each long PCR product were used for library preparation on the Nextera system (Illumina, San Diego, CA, USA), but differences in depth were observed on an amplicon-by-amplicon basis. This phenomenon suggested the possibility that DNA sequencing and the higher order structure of long PCR products could lead to differences in fragmentation efficiency by the Nextera system. In TgCap, repetitive sequences are blocked by Cot1 DNA during probe capture, and this effect was thought to be particularly related to reduced coverage of intron sequences within the target region. Of the 32 samples, 5 samples (ST-15, ST-20, ST-29, ST-56, and ST76) had degraded DNA that could not be subjected to the long PCR, making it impossible to analyze them by vLAS. On the other hand, TgCap could be performed on all samples. There was no difference in variant detection in samples where both vLAS and TgCap were performed.

TML allowed the entire genomic region of each gene of interest to be serially assessed for CpG methylation status. It showed that the CpG sequences of the target genes were mostly methylated, but there was a gap indicating hypomethylation on the 5′ side of the gene (Figure 1, white arrows), which is supposed to corelate with the promoter region. However, the size of the gap varied from gene to gene. It extended not only upstream of exon 1 but also into part of intron 1. The size of the gap is largest in *TP53*, which is a region of about 5 kb continuous with the 5′ upstream of the adjacent *WRAP53* in the inverse direction. *APC* has two gaps at Ex1A and Ex1B. *MSH2* also has two gaps at the boundary region of *EPCAM* and in the region from intron 1 to intron 2 of *MSH2*. Thus, the distribution of hypomethylated regions thought to be associated with promoters varies from gene to gene and may be more widespread than previously thought. As described below, differences in methylation were observed in these regions between normal and tumor tissues, but no other regions that showed clear differences were found. Although vLAS and TgCap are able to detect structural variants such as large deletions, insertions or inversions within genes, all variants detected in tumor DNA were point mutations and no structural variants were observed. Deep intron splice variants were also searched for with SpliceAI, but no variants meeting the criteria were detected.

### 2.3. Variants Detected in Tumor Suppressor Genes

Table 2 and Table 3 summarize the somatic gene variants detected in this study. A more detailed description of the variants is provided in Appendix A. Consistent with the theory that most colorectal cancers begin with biallelic loss-of-function variants of *APC*, *APC* variants were detected in 24 of 32 (75%) tumor DNAs. Most of the variants were frameshift or nonsense (null) variants. *TP53* variants were also detected in 24 of 32 samples, but most were missense variants. In addition, *SMAD4* variants were detected in 7 samples, mostly missense variants. MMR gene variants were found in 10 of 32 samples, and variants in multiple MMR genes were detected simultaneously in two samples (ST-29, ST-76). Of the five samples that were MSI-H by microsatellite analysis, three had MMR gene variants, and the seven with MMR variants were microsatellite stable (MSS). It is unclear why variants in the MMR genes do not necessarily cause MSI-H, but it is possible that mono-allelic variants do not result in LOH, or that microsatellite analysis is missed because it is based on only five markers.

Tumor DNA was obtained from samples taken from areas that a surgeon could visually determine to be cancerous, and a portion of the same specimens was confirmed by a pathologist. However, the pathology and DNA extraction were not performed on the exact same sample, and there is a possibility that normal stromal cells were mixed into the sample, so it cannot be said that 100% of the DNA is of tumor origin. Therefore, if the variant allele frequency (VAF) is greater than 0.5, or if the sum of the VAFs of two different variants found in the same gene (presumed to be in the *trans* position) is greater than 0.5, loss of heterozygosity (LOH) is considered to have occurred in at least a portion of the tumor samples. Based on these criteria, *APC* had LOH in 15 of 24 samples in which variants were detected. For *TP53*, including variants of uncertain significance (VUS), LOH was detected in 10 of 24 samples where variants were detected. However, missense variants of *TP53* can be dominant negative and do not necessarily require LOH [10]. LOH was also observed in 4 of 7 samples with variant detection for *SMAD4*, 1 of 3 for *MSH2*, 2 of 2 for *MSH6*, 1 of 6 for *MLH1*, and 0 of 1 for *PMS2*.

### 2.4. CpG Methylation Alterations in Tumor and Normal Colon Tissue

For each tumor suppressor gene, the TML results were visualized by Integrative Genome Viewer (IGV), and regions with different CpG methylation between normal colon tissue and cancer tissue were extracted (Figure 1 and Figure 2). In *APC*, regions around promoter exons 1A and 1B were hypermethylated in some tumors. The methylation Z-score (MZS) of tumor DNA was more pronounced in exon 1A (Table 2, Figure 2b). For *TP53* and *SMAD4*, some tumors showed hypermethylation from exon 1 upstream to intron 1. However, the degree of hypermethylation was not strong (Table 2, Figure 2c,d). Regarding the MMR genes, hypermethylated regions were found in some tumors in the region between *MSH2* and *EPCAM* and in the region from intron 1 to 2 of *MSH2*, but the degree of hypermethylation was not high when evaluated by MZS (Table 3, Figure 2e,f). *MSH6* showed a hypermethylated region in the region from exon 1 to upstream of intron 1, but the degree of hypermethylation was also not intense (Table 3, Figure 2g). On the other hand, methylation of exon 1 surroundings and upstream region of *MLH1* was found to be very high in degree and frequency (Table 3, Figure 2h). Overall, methylation of key tumor suppressor genes in colorectal cancer was most clearly observed in exon 1A of *APC*, exon 1 surroundings, and upstream regions of *MLH1*. 

### 2.5. Association of Somatic DNA Variants with CpG Methylation in Tumor Suppressor Genes

*APC* hypermethylation was observed in 8 of 32 tumors. *APC* hypermethylation occurred alone in some tumors without DNA variants, such as ST-03 and ST-39, while hypermethylation also occurred in tumors with loss-of-function DNA variants and a high possibility of LOH according to VAF values, such as ST-14, ST-56, and ST-66. Hypermethylation of *TP53* and *SMAD4* was less frequent than that of *APC* and was detected in tumors with no detectable DNA variants or in tumors with low VAF of variants. Regarding the MMR genes, *MLH1* hypermethylation was observed in 10 of 32 tumors, but only 4 of these tumors had MSI-H. Conversely, among the five tumors that showed MSI-H, ST-17 had both *MLH1* variants and hypermethylation, whereas ST-23 and ST-49 had *MLH1* hypermethylation alone, with no variants detected in the MMR genes. ST-29, MSI-H tumor has variants in *MSH2* and *MSH6*, and hypermethylation of *MLH1* has also been detected. ST-57, MSI-H tumor has only a single LOH positive splicing variant of *MLH1* and no hypermethylation of *MLH1* was detected. These results indicate that, in colorectal cancer, somatic DNA variants and CpG hypermethylation of the *APC* and MMR genes are not mutually exclusive but can occur independently and in parallel. It was also shown that hypermethylation of *MLH1* is relatively common (31%) in colorectal cancer, but it is not always associated with MSI-H.

The number of samples corresponding to each class, a classification of different combinations of DNA variants and methylation, is listed in Table 4. It was found that DNA variants and methylation can occur either independently or simultaneously, suggesting that they are essentially independent and parallel processes.

## 3. Discussion

### 3.1. DNA Variants in APC, TP53, SMAD4 and MMR Genes

Consistent with the theory that *APC* is a gatekeeper in colorectal cancer, *APC* somatic DNA variants were detected in 24 of 32 samples (75%), for a total of 41 variants, and LOH was observed in 15 of these samples. In addition, *APC* hypermethylation was observed in eight tumors, five of which were associated with *APC* variants and three of which were not. Including these three cases with hypermethylation alone, somatic *APC* alternations were detected in 27 of the 32 tumors (84.4%). Of the 41 *APC* mutations detected, 34 (82.9%) were frameshift or nonsense (null) variants. *TP53* variants were also detected in 24 of 32 (75%) samples, but, in contrast to *APC*, 19 of the 28 variants (67.8%) were missense variants. The three *TP53* variants detected in the seven samples in this study are commonly reported in other cancers and are known to function as dominant negative variants (Table 2, Appendix A) [23]. Other *TP53* missense variants may also function as dominant negative variants. Three samples showed *TP53* hypermethylation, two of which had no *TP53* variants. Consequently, *APC* and *TP53* show different suppression mechanisms in cancer. In addition to null mutations and loss of function due to LOH, DNA methylation plays an important role in *APC*, whereas TP53 seems to be more likely to select missense variants with dominant-negative effects, which can be expected to have a gene-suppressing effect even with a single hit. A total of eight variants were detected in *SMAD4* in seven samples, seven of which were missense variants. Most of these would be considered VUS according to the guidelines (Table 2). However, four samples were considered LOH according to the VAF value and, therefore, there is a high possibility of pathogenicity. Dominant negative variants of *SMAD4* are not known to be associated with cancer but have been reported in Myhre syndrome (a congenital connective tissue disorder) [24].

MMR gene variants were found in 10 of 32 samples, and variants in multiple MMR genes were detected simultaneously in 2 samples (ST-29, ST-76). Of the five samples that were MSI-H by microsatellite analysis, three had MMR gene variants, and the seven with MMR variants were microsatellite stable (MSS). It is unclear why variants in the MMR genes do not necessarily cause MSI-H, but it is possible that mono-allelic variants do not result in LOH or that microsatellite analysis is missed because it is based on only five markers. In this study, structural abnormalities and intron variants were also screened for using vLAS and TgCap, so it is unlikely that MMR variants exist outside of the exons.

### 3.2. DNA Methylation in Tumor Suppression Genes

TML allowed the entire genomic region of each gene of interest to be serially assessed for CpG methylation status. As expected, in the regions surrounding exon 1 of each gene, gaps were observed indicating hypomethylation of CpG islands. However, the size of the gap varied from gene to gene, and it extended not only upstream of exon 1 but also into part of intron 1 or further. When the methylation status of these gap regions was compared between tumor DNA and normal tissue DNA, it was found that the methylation status of CpG sequences varied in both location and degree from sample to sample and could not be mapped to a specific base sequence. In fact, the DNA methylation status was not consistent even in normal DNA samples. The range and patterns of DNA methylation alterations in cancer are broader and more complex than previously thought. Since both the location of the methylated CpG sequence and the degree of methylation may be related to gene inactivation, in this study, the difference in the degree of methylation between tumor and normal tissues was conveniently expressed using MZS as an index. The gated range for measuring the MZS was set to maximize between tumor and normal DNA.

In several tumor samples, extremely increased methylation was observed around exon 1A of *APC* and exon 1 of *MLH1*. However, methylation patterns were not consistent between tumors, and the extent of methylation tended to be widespread. Other genes showed more modest increased methylation in some tumors. Hypermethylation was observed simultaneously in multiple genes in some samples and in a single gene in others, and it was assumed that methylation of each gene was an essentially independent event. In this study, vLAS and TgCap were used to detect somatic variants. Therefore, splicing abnormalities due to deep intron variants or structural variants were excluded. Therefore, it is highly likely that some tumor samples have only hypermethylation of the target gene as a somatic alternation.

### 3.3. Association of DNA Variants with CpG Methylation in Tumor Suppressor Genes

It is assumed that hypermethylation of exon 1A of *APC* occurs independently and in parallel with somatic variants of *APC*. In some cases (ST-03 and ST-39), methylation occurs alone without *APC* variants; in other cases (ST-14 and ST-56), methylation occurs together with *APC* variants showing LOH; and, in other cases (ST-16 and ST-31), methylation does not occur with *APC* variants showing LOH. Therefore, it was shown that the occurrence of somatic variants and hypermethylation of *APC* can occur in any combination. This suggests that somatic variants and hypermethylation of *APC* are not exclusive events (i.e., if one occurs, the other is suppressed). Furthermore, even if the LOH is achieved due to one event, it is not expected that this will prevent the progress of the other event. Similar phenomena of independence have been observed for *TP53* and *SMAD4*, but to a modest extent. With regard to MMR genes, hypermethylation of *MLH1* occurs at a high frequency independent of somatic variants in MMR genes, including *MLH1*, and does not necessarily coincide with MSI-H. Since microsatellite instability and hypermutation types are a minority in colorectal cancer overall, it is not surprising that the detection rate of somatic variants in the MMR genes was low in this study. However, hypermethylation of *MLH1* was observed in 10 of 32 samples (31.3%), the significance of which is currently unknown.

In any case, the fact that hypermethylation and somatic variants showed independent behavior in all genes examined in this study suggests that DNA methylation is an event that continues to proceed independently of somatic variants in tumor suppressor genes in general.

### 3.4. Limitation of This Study

First, unfortunately, we were unable to obtain RNA from colon tissue in this study, so we could not verify the extent to which high MZS correlates with mRNA expression. In addition, the degree and range of CpG methylation is not constant even in normal colon tissue, making it extremely difficult to derive an appropriate region for the calculation of MZS. Therefore, further studies with larger numbers of samples are needed to determine which patterns and levels of CpG methylation affect mRNA expression and to quantitatively demonstrate the effects. Second, there is no guarantee that normal colon tissue is completely normal. In this study, the surgeons visually separated the normal colon tissue from the cancerous tissue, and a portion of the same sample was confirmed by a pathologist. But, the samples for DNA extraction itself were not pathologically examined, so it cannot be completely ruled out that the seemingly normal tissue may contain cancer cells. For complete separation of normal and cancerous tissue and DNA extraction, it is preferable to use serial sections of formalin-fixed, paraffin-embedded (FFPE) samples; however, this may affect NGS analysis due to reduced DNA quality. TgCap may be possible, but vLAS is not. In TML, cytosine deamination to uracil due to FFPE may interfere with DNA methylation analysis. In addition, the tissue is exposed to inflammatory conditions associated with cancer, which can affect DNA methylation status. In fact, *TP53* methylation is known to occur in various pathological conditions other than cancer, such as ischemic stroke [25] and asthma [26]. In addition, the level of methylation may be influenced by individual differences in the activity of genes involved in DNA methylation, such as DNA methyltransferase or ten-eleven translocation (TET)-mediated DNA demethylation. It is also known that lifestyle, including diet, and, therefore, the presence of certain molecules or potential contaminants, can directly or indirectly contribute to the status of DNA methylation [27,28,29]. There have also been reports that microbiota, which refers to the composition and abundance of microorganisms, affects DNA methylation in colorectal cancer [30]. As shown, the factors involved in DNA methylation are extremely diverse and it is extremely difficult to standardize them across patients. In this study, samples were obtained from resected tissue after surgery, but to obtain high quality RNA, the method of sample collection itself required ingenuity. In addition, if the target gene has a somatic null variant, the amount of mRNA may be reduced due to nonsense-mediated mRNA decay. Also, methylation of *APC* exons 1A and 1B may affect different classes of mRNA variants. To distinguish these factors from the effect of CpG hypermethylation, it is necessary to choose an analysis method such as targeted RNA-seq [31] or allele-separated RNA-seq with long-read sequencing [32]. This study demonstrated that DNA methylation of tumor suppressor genes in colorectal cancer is more extensive and irregular than previously thought and occurs in parallel with somatic variants. However, to evaluate its effect on mRNA expression and gene silencing, it will be necessary to overcome the above discussion.

## 4. Materials and Methods

### 4.1. Patient and Sample Collection

The patients in this study are clinically diagnosed with colorectal cancer at Kanazawa Medical University Hospital. Patients with suspected familial adenomatous polyposis, Lynch syndrome or Li-Fraumeni syndrome based on family history, medical history or preoperative examination findings were excluded. Clinical data collected included age at surgery, sex, site of colorectal cancer and results of postoperative pathological testing for *KRAS*, *BRAF* variants and microsatellite instability (MSI). *KRAS* variants at codons 12, 13, 59, 61, 117, and 146 and *BRAF* variants at codon 600 were analyzed, and MSI was performed using BAT-26, NR-21, BAT-25, MONO-27, and NR-24 markers. These analyses were performed in a clinically certified laboratory (SRL Co., Ltd., Tokyo, Japan). Other analyses in this study used surgically resected, unfixed bowel tissue. After surgery, cancerous and normal bowel tissue were collected separately as paired samples under macroscopic examination by a gastroenterologist. A portion of the same specimens were fixed in formalin and confirmed by a pathologist to be either cancerous or normal tissue. Written informed consent was obtained from all patients by their primary physician for the mutational and clinical data analysis. The study design was approved by the institutional review board of Kanazawa Medical University (I544, G186).

### 4.2. Library Preparations and Sequencing

Standard phenol/chloroform extraction and ethanol precipitation was used to extract DNA from colon tissue samples. In this study, three different libraries were generated from each sample to comprehensively investigate seven major tumor suppressor genes (*APC*, *TP53*, *SMAD4*, *MSH2*, *MSH6*, *MLH1*, *PMS2*) in colorectal cancer using next-generation sequencing (Figure 3).

Three different libraries were generated from tumor DNA and normal tissue DNA and analyzed simultaneously by NGS. The first library is our previously developed method, called the very long amplicon sequencing (vLAS) [33,34,35]. vLAS is a targeted DNA-Seq based on long range PCR-based NGS, and, by determining the DNA sequence of the entire gene region, including the intron, it is possible to detect not only point mutations in exons and their surrounding areas, but also structural variants such as large deletions and splicing abnormalities due to deep intron variants. In this method, the extracted DNA is amplified by a 20–30 kb long range PCR, the target region, including intron sequences, is covered with multiple amplicons, and a library is then prepared using a Nextera system (Illumina, San Diego, CA, USA). A list of the long PCR primers used in this study is provided in Appendix A. This method has the advantage of being able to detect structural abnormalities such as intragenic deletions and deep intron mutations in addition to point mutations, but it has the disadvantage of not being able to be used with low molecular weight DNA because long PCR cannot be performed.

Targeted genome capture sequencing (TgCap) has therefore been developed to enable analysis of the entire target genomic region similar to vLAS, even with low molecular weight DNA. TgCap is a capture sequencing method that uses biotin-labeled probes generated from fragments of long PCR products. It can be used even when the sample DNA is degraded and cannot be amplified by long PCR. In TgCap, the long PCR products amplified from control DNA using vLAS primers are fragmented by ultrasound and biotin end-labeling to create a capture probe. First, each long PCR product was quantified using Qubit (Thermo Fisher Scientific, Waltham, MA, USA) and mixed in equimolar amounts. The mixed products were then fragmented to 300 bp using Covaris M220 (Covaris. Wouburn, MA, USA). One microgram of fragmented DNA was treated with the Fast DNA End Repair Kit (Thermo Fisher Scientific), followed by terminal transferase (New England Biolabs, Ipswich, MA, USA) labeling with a final 0.1 mM biotin 16-ddUTP (New England Biolabs) at 37 °C for 2 h. The final probe concentration was adjusted to 10 ng/μL. Sample genomic DNA was fragmented to 300 bp using Covaris M220 and treated with NEBNext^®^ Ultra™ II End Repair/dA-Tailing Module (New England Biolabs). Then, 500 ng of DNA and 75 pmol xGen UDI-UMI Adapter (Integrated DNA Technology, Coralville, IA, USA) were ligated with Blunt TA ligase (New England Biolabs) at 20 °C for 1 h. This genomic DNA library was captured with 1 μL of the prepared biotin probe using a KAPA Hyper Capture Reagent Kit (Roche Diagnostics, Basel, Switzerland) according to the manufacturer’s protocol. Finally, the captured DNA was amplified using KAPA HiFi HotStart ReadyMix (Roche Diagnostics) to obtain a library. In this study, we excluded *PMS2* from TgCap because the presence of pseudogenes poses a capture problem. In vLAS, the use of *PMS2* specific PCR primers allowed for analysis without pseudogene contamination.

Targeted methyl landscape (TML) is a method to perform targeted methyl-Seq on the entire region covered by TgCap. Therfore, TML is an application of TgCap in which methylated adapters are used for capture in the same way as TgCap and the captured DNA is then bisulfite treated and amplified to produce a methylation library of the entire target region. The xGen Methyl UDI-UMI Adapter (Integrated DNA Technology) was used as the methylation adapter and the EZ DNA Methylation Lightining Kit (Zymo Research, Irvine, CA, USA) was used for the bisulphite treatment. Multiplexed capture of up to four samples using different indices was possible with both TgCap and TML.

Libraries were quantified using an HS Qubit dsDNA assay (Thermo Fisher Scientific) and a TapeStation 4200 (Agilent Technologies, Santa Clara, CA, USA). Qualified size distributions were checked on a TapeStation 4200 using High Sensitivity D1000 ScreenTape (Agilent Technologies). A 12.5 pM library was sequenced on an Illumina MiSeq system (2 × 150 cycles or 2 × 250 cycles), according to the standard Illumina protocol (Illumina). For a more detailed protocol, please contact the corresponding author (Y.N.).

### 4.3. DATA Analysis Pipeline

The FASTQ files were generated using the bcl2fastq software v1.8.4 (Illumina). The FASTQ files were aligned to the reference human genome (hg38) using the Burrows-Wheeler Aligner MEM algorithm (BWA-MEM version 0.7.17-r1188) [36]. Somatic variants were identified using GATK’s Mutect2 (Version 4.0.6.0) [37,38,39]. The SNVs and INDELs were functionally annotated by SnpEff (Version 4.3t), to classify each variant into a functional class (HIGH, MODERATE, LOW, and MODIFIER) [40]. For variant annotation, the Database of Short Genetic Variations dbSNP (Version 151) and ClinVar were used [41,42]. To detect large INDELs, Pindel (Version 0.2.5b9) was used [43]. Bismark (version v0.22.1) was used as the methylation caller for bisulfite sequencing [44]. Splice variants, including deep intronic variants, were searched for using SpliceAI [45] and were considered significant if either donor gain/loss or acceptor gain/loss values were >0.5 and the allele frequency in the general population was less than 0.0001. For visualization, the Integrative Genomic Viewer (IGV Version 2.4.13) was used [46] (Figure 1).

### 4.4. Annotation of Variants

The oncogenicity of detected variants was determined according to the joint recommendations of the Clinical Genome Resource (ClinGen), Cancer Genomics Consortium (CGC), and Variant Interpretation for Cancer Consortium (VICC) [47]. This guideline classifies oncogenicity into five categories, benign (−7 points or less), likely benign (−1 to −6 points), variant of uncertain significance (VUS) (0 to 5 points), likely oncogenic (6 to 9 points), and oncogenic (10 points or more). Since a null variant in a tumor suppressor gene alone would score 8 points, in this study, nonsense variants, frameshift variants, and splicing variants were automatically considered likely oncogenic or oncogenic, where pathogenicity is obvious. Therefore, missense and in-frame variants were classified according to this guideline (Table 2 and Table 3).

### 4.5. Evaluation of CpG Methylation

When the results of the Bismark analysis were visualized using IGV, it was found that CpG methylation was not concentrated at specific bases, but randomly distributed over a certain area in both normal colon tissue and colon cancer tissue. In this study, we developed the Methylation Z-Score (MZS) to quantify the degree of CpG methylation in the target region. The ratio of CpG methylation at each base was calculated by Bismark and visualized by loading the bedGraph into IGV. Extract regions on IGV that have different methylation levels between normal and cancer tissues. The grand sum of the CpG methylation ratios (0–1) for each base within the gated region was calculated for 32 normal colon tissue samples, and the average (av.ND_Σ) and standard deviation (SD) were calculated. The grand sum was calculated similarly for each colon cancer samples (TD_Σ) and the methylation level was expressed as [(TD_Σ) − (av.ND_Σ)]/SD = MZS. The gating region has been adjusted for maximum MZS. In addition, MZS was classified as follows: 2 to 5 SD as high (H), 5 to 10 SD as very high (VH), and greater than 10 SD as extremely high (EH). The degree of DNA methylation of each tumor suppressor gene in each cancer sample is shown by this classification in Table 2 and Table 3, and by the actual MZS value in Figure 1.

## 5. Conclusions

Using vLAS, TgCap and TML, we were able to perform an integrated analysis of somatic variants and CpG methylation status of major tumor suppressor genes in colorectal cancer. vLAS was found to have the advantage of more uniform coverage, while TgCap was found to have the advantage of being applicable to degraded DNA. TML enabled analysis of the CpG methylation status of tumor suppressor genes in colorectal cancer across all gene regions, demonstrating that DNA hypermethylation in cancer is more complex and widespread than previously thought. Somatic variants and hypermethylation of tumor suppressor genes were considered independent, parallel events. To understand the true impact of this complex hypermethylation on the mRNA and protein expression of tumor suppressor genes, further detailed research with larger sample sizes and optimal sample processing is needed. In addition to the NGS analysis methods demonstrated in this study, this research will require comprehensive analysis using multiple methods, including RNA-seq, immunohistochemistry, and western blotting.

## Figures and Tables

**Figure 1 ijms-26-01642-f001:**
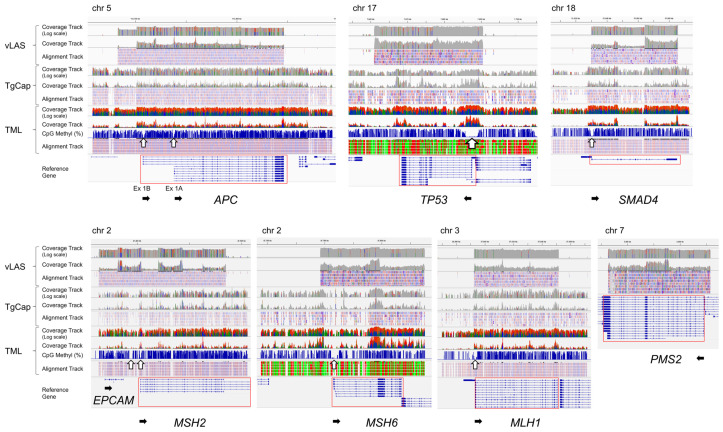
An example of an IGV image of vLAS, TgCap and TML (ST-57 tumor DNA). The CpG methylations shown in the TML graph have gaps corresponding to the promoter regions of each gene (white arrows), indicating hypomethylation of CpG islands. The black arrows show the orientation of each gene on the genome.

**Figure 2 ijms-26-01642-f002:**
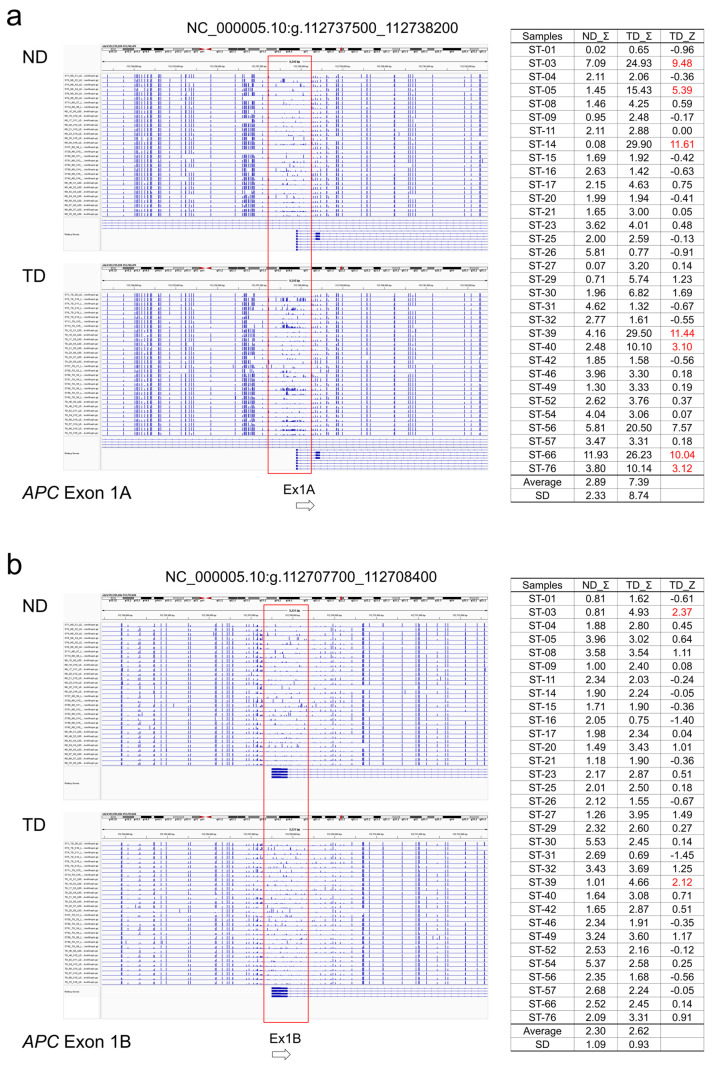
IGV image of TML CpG methylation ratio indicated by Bismark bedGraph. (**a**) *APC* promotor exon 1A. (**b**) *APC* promotor exon 1B. ND; Normal colon tissue DNA, ND_Σ; Sum of the CpG methylation ratios of ND in the target region, TD; colon tumor DNA, TD_Σ; Sum of the CpG methylation ratios of TD in the target region, TD_Z; Methylation Z score of TD. (**c**) *TP53* exon 1~intron 1. (**d**) *SMAD4* exon 1~intron 1. (**e**) *EPCAM MSH2* intervening region. (**f**) *MSH2* intron 1~2. (**g**) *MSH6* exon 1~intron 1. (**h**) *MLH1* 5′UTR~exon 1.

**Figure 3 ijms-26-01642-f003:**
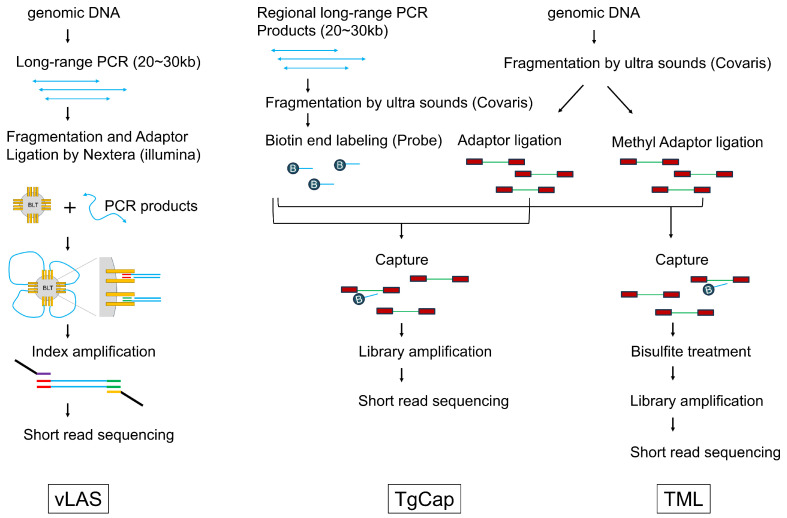
Library preparation methods of vLAS, TgCap and TML.

**Table 1 ijms-26-01642-t001:** Clinical profile of the patients with colorectal cancer.

Patient	Age	Sex	Locus	*KRAS*	*BRAF*	MSI
ST-01	62	F	T	-	-	MSS
ST-03	79	M	R	G12D	-	MSS
ST-04	55	F	S	G13D	-	MSS
ST-05	90	F	T	-	-	MSS
ST-08	86	F	C	-	-	MSS
ST-09	73	M	A	G12D	-	MSS
ST-11	67	M	R	-	-	MSS
ST-14	67	F	S	G12D	-	MSS
ST-15	80	M	C	G13D	-	MSS
ST-16	70	M	D	-	-	MSS
ST-17	84	M	A	-	V600E	MSI-H
ST-20	81	F	C	G13D	-	MSS
ST-21	64	F	S	-	-	MSS
ST-23	85	F	A	-	V600E	MSI-H
ST-25	64	M	R	-	-	MSS
ST-26	74	F	A	-	-	MSS
ST-27	77	M	D	G12D	-	MSS
ST-29	74	M	AR	-	-	MSI-H
ST-30	61	F	A	G13C	-	MSS
ST-31	84	M	A	G12V	-	MSS
ST-32	55	F	A	G12D	-	MSS
ST-39	73	F	T	-	-	MSS
ST-40	85	M	R	-	V600E	MSS
ST-42	83	F	C	G13C	-	MSS
ST-46	68	M	D	G12D	-	MSS
ST-49	82	F	A	-	V600E	MSI-H
ST-52	88	M	S	-	-	MSS
ST-54	75	M	S + T(LST)	-	-	MSS
ST-56	70	M	S + ML	-	-	MSS
ST-57	48	F	R	-	-	MSI-H
ST-66	64	F	R + LM	-	-	MSS
ST-76	59	M	S	-	-	MSS

A: ascending colon, C: cecal, D: descending colon, LM: liver metastasis, LST: laterally spreading tumor, ML: malignant lymphoma, R: rectal, S: sigmoid, T: transverse colon, MSI: microsatellite instability, MSI-H: MSI High, MSS: microsatellite stable.

**Table 2 ijms-26-01642-t002:** Somatic variants and methylation profile 1 (*APC*, *TP53*, *SMAD4*).

Tumor	*APC*				*TP53*			*SMAD4*		
DNA	Variants	LOH	MZS	MZS	Variants	LOH	MZS	Variants	LOH	MZS
Sample	(VAF)		Ex1A	Ex1B	(VAF)			(VAF)		
ST-01	R252* (0.33) N1818fs (0.40)	Y	-	-	R209Q (0.13) ^LO,DN^	N	-	-		-
ST-03	-		VH	H	Y181N (0.25) ^V^	N	-	-		-
ST-04	E763* (0.34)	N	-	-	R234H (0.51) ^LO,DN^	Y	-	-		-
ST-05	R71C (0.10) ^V^ S457* (0.09) T1556fs (0.04)	N	H	-	-		H	-		-
ST-08	S943* (0.59) P1497fs (0.22) ^C^ T1556fs (0.20) ^C^	Y	-	-	R209Q (0.67) ^LO,DN^	Y	-	N64fs (0.66)	Y	-
ST-09	T1368fs (0.24) S1415fs (0.42)	Y	-	-	-		-	-		-
ST-11	Y935fs (0.44)	N	-	-	S176C (0.09) ^V^	N	-	R361C (0.12) ^LO^	N	-
ST-14	V452fs (0.24) E1237* (0.34)	Y	EH	-	E246G (0.31) ^V^	N	-	-		-
ST-15	E1397* (0.52)	Y	-	-	Y387fs (0.17) Q97E (0.23) ^V^	N	-	-		-
ST-16	Q1367* (0.95)	Y	-	-	E255fs (0.91)	Y	-	-		-
ST-17	L1449N (0.40) ^V^	N	-	-	-		H	-		-
ST-20	Ser1539* (0.92)	Y	-	-	-		-	-		-
ST-21	-		-	-	c.-32delA (0.44)	N	-	-		-
ST-23	-		-	-	V118D (0.82) ^V^	Y	-	C363R (0.77) ^V^	Y	-
ST-25	G635fs (0.15)	N	-	-	Ile123fs (0.06)	N	-	-		-
ST-26	S245* (0.62) S2146L (0.15) ^V^	Y	-	-	G206S (0.88) ^LO^	Y	-	-		-
ST-27	1357* (0.33)	N	-	-	E255G (0.33) ^V^	N	-	-		-
ST-29	S1163_Y1166del (0.13) E1265fs (0.23) S2295_R2301del (0.10) R2347fs (0.11) S2512fs (0.11)	N	-	-	L383fs(0.54) R243W (0.14) ^LO,DN^	Y	H	-		-
ST-30	S874* (0.35) R1435* (0.31)	Y	-	-	R234H (0.49) ^T,DN^ R228W (0.12) ^T^	Y	-	-		H
ST-31	R876* (0.63) R1450* (0.17)	Y	-	-	R209Q (0.86) ^LO,DN^	Y	-	-		-
ST-32	R1314fs (0.91)	Y	-	-	-		-	-		-
ST-39	-		EH	H	V118D (0.73) ^V^	Y	-	C363R (0.81) ^V^	Y	-
ST-40	-		H	-	V134L (0.43) ^LO^	N	-	A118V (0.35) ^V^	N	-
ST-42	E901* (0.30) T1556fs (0.18)	N	-	-	Q126* (0.22)	N	-	-		-
ST-46	D1394fs (0.43)	N	-	-	c.258G>A T86 = (0.27) ^Sp^	N	-	-		-
ST-49	-		-	-	-		-	-		-
ST-52	V1405fs (0.63)	Y	-	-	-		-	-		-
ST-54	-		-	-	E17* (0.31)	N	-	-		-
ST-56	S1421fs (0.59) S2607F (0.29) ^V^	Y	VH	-	G227E (0.50) ^V^	Y	-	L495H (0.33) ^V^	N	-
ST-57	-		-	-	-		-	-		-
ST-66	S1315fs (0.67)	Y	EH	-	C102Y (0.45) ^V^	N	-	-		-
ST-76	c.835-8A>G (0.60) ^Sp^ Q2742* (0.25)	Y	H	-	Y387fs (0.645) R209Q (0.50) ^LO,DN^	Y	-	C401Y (0.25) ^V^ A456V (0.40) ^V^	Y	-

EH: extremely high (≦10 SD), H: high (2~5 SD), LOH: loss of heterozygosity, MZS: methylation Z-score, N: No LOH, VAF: variant allele frequency, VH: very high (5~10 SD), Y: LOH positive, ^C^: cis, ^DN^: Dominant negative variant, ^LO^: Likely oncogenic, ^T^: trans, ^Sp^: known splicing variant, ^V^: variant of uncertain significance (VUS).

**Table 3 ijms-26-01642-t003:** Somatic variants and methylation profile 2 (MMR genes).

Tumor	*MSH2*				*MSH6*			*MLH1*			*PMS2*
DNA	Variants	LOH	MZS	MZS	Variants	LOH	MZS	Variants	LOH	MZS	Variants
Sample	(VAF)		5′UTR	In1_2	(VAF)			(VAF)			(VAF)
ST-01	-		-	-	-		-	-		H	-
ST-03	-		-	-	-		-	-		EH	M676V (0.42) ^V^
ST-04	-		-	-	-		-	-		-	-
ST-05	-		-	-	-		-	-		EH	-
ST-08	-		-	-	-		-	-		-	-
ST-09	Q893* (0.18)	N	-	-	-		-	-		-	-
ST-11	-		-	-	-		-	-		-	-
ST-14	-		-	-	-		-	-		EH	-
ST-15	-		-	-	-		-	-		-	-
ST-16	-		-	-	-		-	-		-	-
ST-17	-		-	-	-		-	N444fs (0.34)	N	EH	-
ST-20	-		-	-	-		-	1731+2T>C (0.25)	N	-	-
ST-21	-		-	-	-		-	-		-	-
ST-23	-		-	-	-		-	-		EH	-
ST-25	-		-	-	-		-	Q197H (0.10) ^V^	N	-	-
ST-26	-		-	-	-		-	F656C (0.09) ^V^	N	-	-
ST-27	-		H	H	-		H	-		-	-
ST-29	Q4* (0.25)	N	-	-	S625C (0.27) ^V^ I1170fs (0.25)	Y	-	-		EH	-
ST-30	-		-	-	-		-	-		-	-
ST-31	-		-	-	-		H	-		-	-
ST-32	-		H	H	-		H	-		-	-
ST-39	-		-	H	-		-	-		-	-
ST-40	-		H	H	-		H	-		-	-
ST-42	-		H	-	-		H	-		-	-
ST-46	-		-	-	-		-	-		-	-
ST-49	-		-	-	-		-	-		VH	-
ST-52	-		-	-	-		-	-		EH	-
ST-54	-		-	-	-		-	-		-	-
ST-56	-		-	-	-		-	M587I (0.25) ^V^	N	H	-
ST-57	-		-	-	-		-	c.306+1G>A (0.60)	Y	-	-
ST-66	-		H	-	-		-	-		-	-
ST-76	N919D (0.50) ^V^	Y	-	-	E807* (0.40) K1315R (0.50) ^V^	Y	-	-		-	-

EH: extremely high, H: high, LOH: loss of heterozygosity, MMR: mismatch repair, MZS: methylation Z-score, N: No LOH, VAF: variant allele frequency, VH: very high, Y: LOH positive, ^V^: variant of uncertain significance (VUS).

**Table 4 ijms-26-01642-t004:** Summary of somatic variants and hyper-methylation of 32 colorectal cancers.

Class	2nd Hit			Gene						Total
	Variant	LOH	HM	*APC*	*TP53*	*SMAD4*	*MSH2*	*MSH6*	*MLH1*	
A	N	N	N	5	6	24	24	25	18	102
B	Y	N	N	8	13	3	2	0	3	28
C	Y	Y	N	11	10	4	1	2	1	30
D	Y	N	Y	1	0	0	0	0	2	3
E	Y	Y	Y	4	1	0	0	0	0	5
F	N	N	Y	3	2	1	5	5	8	24

HM: Hyper-methylation, LOH: loss of heterozygosity, N: no (negative), Y: yes (positive).

## Data Availability

The NGS sequencing data presented in this study are available upon request from the corresponding author (Y.N.), as they are subject to disclosure restrictions under the Japanese government’s Personal Information Protection Act and consent of the subjects was not obtained.

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
