# Peer review of "Integrated Analysis of Somatic DNA Variants and DNA Methylation of Tumor Suppressor Genes in Colorectal Cancer"

_ijms, 2025, doi:10.3390/ijms26041642_

Round 1
Reviewer 1 Report
Comments and Suggestions for Authors
The authors performed a comprehensive analysis of somatic variants and DNA methylation status of major oncogenes (e.g., APC, TP53, SMAD4, and MMR genes) in colorectal cancer by means of novel next-generation sequencing technologies (vLAS, TgCap, TML). The article is novel and scientific in its approach, but there is still room for improvement, especially in validating the functional significance of methylation and improving the sample selection criteria. It is recommended that the revision be accepted for publication.
1. The transition to the research methodology section directly after the introduction is slightly abrupt. A succinct summary paragraph could be included in the final section to clarify how this study addresses the shortcomings in existing research through technological innovation and to pave the way for the next step in the research methodology.
2. The roles of DNA methylation and somatic mutations in colorectal cancer are described in the introduction, but it could be further expanded to explain the mechanisms by which both are involved in carcinogenesis and how they synergistically affect the function of oncogenes. Appropriate references to relevant literature could be added to show the known functions of oncogenes (e.g., APC, TP53, etc.) in cancer and the disruption of their oncogenic effects by mutation or methylation.
3. In the introduction, although the effects of DNA methylation and somatic variants on oncogenes are mentioned, the lack of existing studies and the innovative nature of the present study are not sufficiently emphasized. The innovation of this study could be highlighted by pointing out more clearly the lack of understanding of independent studies of oncogene methylation and somatic variants and their interactions in the current literature.
4. The presentation of some results (e.g., the methylation distribution graph in Figure 1) was not clear enough, and the graph annotation and color contrast could be further optimized to improve readability.
5. Normal tissues may be affected by cancer or inflammation and additional pathology verification is recommended to ensure sample purity.
6. The current sample size of 32 cases is a small coverage that may affect the generalizability of the results, or even if benign patients could be added? Future studies may increase the sample size to verify the broadness of the findings.
7. Some of the conclusions (e.g., independence of methylation from variation) are based on observations, and it is recommended that more detailed statistical tests be provided to enhance persuasiveness.
Comments on the Quality of English LanguageThe English could be improved to more clearly express the research.
Author Response
Response to Reviewer 1
Dear Reviewer 1,
Thank you for giving us the opportunity to submit a revised draft of our manuscript to the International Journal of Molecular Science. We appreciate the time and effort that you have dedicated to providing your valuable feedback on our manuscript. We are grateful to you for your insightful comments on our paper. We have been able to incorporate changes to reflect all of your suggestions. In revised manuscript, we indicated the changes with Highlight.
Here is a point-by point response to your comments and concerns.
Q1. The transition to the research methodology section directly after the introduction is slightly abrupt. A succinct summary paragraph could be included in the final section to clarify how this study addresses the shortcomings in existing research through technological innovation and to pave the way for the next step in the research methodology.
A1. A similar comment was made by reviewer 2. The analysis methodology has been moved to Section 4.2 Library Preparation and Sequencing.
Q2. The roles of DNA methylation and somatic mutations in colorectal cancer are described in the introduction, but it could be further expanded to explain the mechanisms by which both are involved in carcinogenesis and how they synergistically affect the function of oncogenes. Appropriate references to relevant literature could be added to show the known functions of oncogenes (e.g., APC, TP53, etc.) in cancer and the disruption of their oncogenic effects by mutation or methylation.
A2. The introduction has been completely revised in line with your suggestions. We have summarized basic information about the two-hit theory of tumor suppressor genes and DNA methylation in colorectal cancer and added references. See line 31-43.
Q3. In the introduction, although the effects of DNA methylation and somatic variants on oncogenes are mentioned, the lack of existing studies and the innovative nature of the present study are not sufficiently emphasized. The innovation of this study could be highlighted by pointing out more clearly the lack of understanding of independent studies of oncogene methylation and somatic variants and their interactions in the current literature.
A3. We explained the content of existing research by citing papers and emphasizing the importance of our research. See line 46-54.
Q4. The presentation of some results (e.g., the methylation distribution graph in Figure 1) was not clear enough, and the graph annotation and color contrast could be further optimized to improve readability.
A4. Figure 1 has been revised and replaced for clarity. Figure 2 has had its contrast adjusted.
Q5. Normal tissues may be affected by cancer or inflammation and additional pathology verification is recommended to ensure sample purity.
A5. In fact, a portion of the tissue from which the DNA was extracted is fixed in formalin and examined by a pathologist. However, it is important to note that the tissue is not exactly the same as that described in lines 150-152, 331-338, and 375-377.
Q6. The current sample size of 32 cases is a small coverage that may affect the generalizability of the results, or even if benign patients could be added? Future studies may increase the sample size to verify the broadness of the findings.
A6. This is a pilot study and we plan to conduct more detailed studies in the future. See line 496-501.
Q7. Some of the conclusions (e.g., independence of methylation from variation) are based on observations, and it is recommended that more detailed statistical tests be provided to enhance persuasiveness.
A7. Due to the small sample size, it is difficult to draw any statistical conclusions, so we have instead presented the results in Table 4 to aid the reader's understanding.
Through this revision, we feel that we could improve our article. Thank you very much for your valuable suggestions.
Sincerely,
Yo Niida, M.D., Ph.D.
Center for Clinical Genomics
Kanazawa Medical University Hospital
- Daigaku, Uchinada, Kahoku, Ishikawa, 920-0293, JAPAN
Phone No: +81 076-286-2211
Email Address: niida@kanazawa-med.ac.jp

Reviewer 2 Report
Comments and Suggestions for Authors
In the introduction, the authors discussed extensively the novel approaches using next-generation sequencing (NGS) to simultaneously analyze somatic variants and DNA methylation as vLAS (very long amplicon sequencing), TgCap (targeted genome 63 capture), and TML (targeted methyl landscape) that they used in their work. This description is better to be mentioned in the methodology section not the introduction and the authors can enrich the introduction with more information about somatic DNA variants and DNA methylation of tumor suppressor genes in colorectal cancer.
Author Response
Response to Reviewer 2
Dear Reviewer 2,
Thank you for giving us the opportunity to submit a revised draft of our manuscript to the International Journal of Molecular Science. We appreciate the time and effort that you have dedicated to providing your valuable feedback on our manuscript. We are grateful to you for your insightful comments on our paper. We have been able to incorporate changes to reflect your suggestions. In revised manuscript, we indicated the changes with Highlight.
Here is a point-by point response to your comments and concerns.
Q1. In the introduction, the authors discussed extensively the novel approaches using next-generation sequencing (NGS) to simultaneously analyze somatic variants and DNA methylation as vLAS (very long amplicon sequencing), TgCap(targeted genome 63 capture), and TML (targeted methyl landscape) that they used in their work. This description is better to be mentioned in the methodology section not the introduction and the authors can enrich the introduction with more information about somatic DNA variants and DNA methylation of tumor suppressor genes in colorectal cancer.
A1. The introduction has been completely revised in line with your suggestions. The analysis methodology has been moved to Section 4.2 Library Preparation and Sequencing. We have summarized basic information about the two-hit theory of tumor suppressor genes and DNA methylation in colorectal cancer and added references.
Through this revision, we feel that we could improve our article. Thank you very much for your valuable suggestions.
Sincerely,
Yo Niida, M.D., Ph.D.
Center for Clinical Genomics
Kanazawa Medical University Hospital
- Daigaku, Uchinada, Kahoku, Ishikawa, 920-0293, JAPAN
Phone No: +81 076-286-2211
Email Address: niida@kanazawa-med.ac.jp

Reviewer 3 Report
Comments and Suggestions for Authors
In this article, the authors provide a comparative analysis of DNA extracted from colorectal cancer tissue and normal colon tissue obtained from surgical specimens from 32 patients for APC, TP53, SMAD4, MSH2, MSH6, MLH1, and PMS2 genes using new NGS techniques. The manuscript is straightforward, well written, and concise and has clear results. Definitely deserves to be published and is a valuable contribution to the “International Journal of Molecular Sciences” journal.
Please, address the following comment, as per recommended.
“1. Introduction”, Page 2 of 20, Lines 47-50:
“On the other hand, inactivation of mismatch repair (MMR) genes (MSH2, MSH6, MLH1, and PMS2) is associated with microsatellite instability and hypermutation (13%), and germline variants in MMR genes cause Lynch syndrome (hereditary nonpolyposis colon cancer)[11].”.
At that point, the authors should mention that immune cell PD-L1 expression is significantly higher in MMR-deficient (MSI-H) colorectal cancer as compared to MMR-proficient (MSI-L) tumors, with no differences among the different MSI-H molecular subtypes. The recommended screening for defective, DNA MMR includes immunohistochemistry (IHC) and/or MSI test. However, there are challenges in distilling the biological and technical heterogeneity of MSI testing down to usable data. It has been reported in the literature that IHC testing of the MMR machinery may give different results for a given germline mutation and has been suggested that this may be due to somatic mutations.
Recommended reference: Adeleke S, et al. Microsatellite instability testing in colorectal patients with Lynch syndrome: lessons learned from a case report and how to avoid such pitfalls. Per Med. 2022;19(4):277-286.
Author Response
Response to Reviewer 3
Dear Reviewer 3,
Thank you for giving us the opportunity to submit a revised draft of our manuscript to the International Journal of Molecular Science. We appreciate the time and effort that you have dedicated to providing your valuable feedback on our manuscript. We are grateful to you for your insightful comments on our paper. We have been able to incorporate changes to reflect your suggestions. In revised manuscript, we indicated the changes with Highlight.
Here is a point-by point response to your comments and concerns.
Q1. Please, address the following comment, as per recommended.
“1. Introduction”, Page 2 of 20, Lines 47-50:
“On the other hand, inactivation of mismatch repair (MMR) genes (MSH2,MSH6, MLH1, and PMS2) is associated with microsatellite instability and hypermutation (13%), and germline variants in MMR genes cause Lynch syndrome (hereditary nonpolyposis colon cancer)[11].”.
At that point, the authors should mention that immune cell PD-L1 expression is significantly higher in MMR-deficient (MSI-H) colorectal cancer as compared to MMR-proficient (MSI-L) tumors, with no differences among the different MSI-H molecular subtypes. The recommended screening for defective, DNA MMR includes immunohistochemistry (IHC) and/or MSI test. However, there are challenges in distilling the biological and technical heterogeneity of MSI testing down to usable data. It has been reported in the literature that IHC testing of the MMR machinery may give different results for a given germline mutation and has been suggested that this may be due to somatic mutations.
Recommended reference: Adeleke S, et al. Microsatellite instability testing in colorectal patients with Lynch syndrome: lessons learned from a case report and how to avoid such pitfalls. Per Med. 2022;19(4):277-286.
A1. The introduction has been revised in line with your suggestions. WE add next sentences in line 70-77, and add the reference you recommended.
MSI-H or mismatch repair-deficient (dMMR) colorectal cancer is characterized by high expression of PD-L1 (programmed death ligand-1) and is therefore amenable to treatment with anti-PD-1 (programmed cell death protein-1)/PD-L1 checkpoint inhibitors , a type of immunomodulatory therapy[21]. Immunohistochemistry for dMMR or MSI testing of DNA is recommended for evaluation of MMR deficiency, but can be difficult to interpret in some cases. It has been reported that dMMR results may not be consistent with germline mutations, and it has been suggested that the cause may depend on the status of somatic mutations[22].
Through this revision, we feel that we could improve our article. Thank you very much for your valuable suggestions.
Sincerely,
Yo Niida, M.D., Ph.D.
Center for Clinical Genomics
Kanazawa Medical University Hospital
- Daigaku, Uchinada, Kahoku, Ishikawa, 920-0293, JAPAN
Phone No: +81 076-286-2211
Email Address: niida@kanazawa-med.ac.jp

Reviewer 4 Report
Comments and Suggestions for Authors
The manuscript sounds good. I have some suggestions to improve it.
Lines 224- 228 The discussion part about DNA variants in APC is not really clear. I suggest to better clarify this part.
Lines 273-275 The authors asserted that “methylation of each gene was an essentially independent event”, this comes out of the investigations carried out on the patients in question affected by colorectal cancer. It is important to underline how, given the different context in which cells can find themselves, for example inflammatory stage, their genic and genomic methylation status can also be influenced by something else that is not strictly linked to the oncological condition, but also simply to a variety in the activity of DNMTs and TETs for example. At the same time, it is also known in the literature (PMID: 36635363; PMID: 37571432; PMID: 38612845) that lifestyle, for example food, and therefore the presence of specific molecules, or potential pollutants in water, can directly or indirectly contribute to the status of genetic and genomic methylation, collaborating with the SAM-DNMT complex; this is the reason why methylation status and also its variations can be considered multifactorial dependent. We therefore recommend to more discuss this aspect in the text, also by inserting the above-indicated references.
Line 445 It would be better to check protein expression via WB.
Author Response
Response to Reviewer 4
Dear Reviewer 4,
Thank you for giving us the opportunity to submit a revised draft of our manuscript to the International Journal of Molecular Science. We appreciate the time and effort that you have dedicated to providing your valuable feedback on our manuscript. We are grateful to you for your insightful comments on our paper. We have been able to incorporate changes to reflect all of your suggestions. In revised manuscript, we indicated the changes with Highlight.
Here is a point-by point response to your comments and concerns.
Q1. Lines 224- 228 The discussion part about DNA variants in APC is not really clear. I suggest to better clarify this.
A1. This section has been rewritten as follows in line 241-247:
Consistent with the theory that APC is a gatekeeper in colorectal cancer, APC somatic DNA variants were detected in 24 of 32 samples (75%), for a total of 41 variants, and LOH was observed in 15 of these samples. In addition, APC hypermethylation was observed in eight tumors, five of which were associated with APC variants and three of which were not. Including these three cases with hypermethylation alone, somatic APC alternations were detected in 27 of the 32 tumors (84.4%). Of the 42 APC mutations detected, 34 (82.9%) were frameshift or nonsense (null) variants.
The total number of variants was incorrectly reported and has been corrected from 42 to 41. For the reader's convenience, we have added LOH columns to Tables 2 and 3. We have also added a list of valiant and methylation distributions to Table 4.
Q2. Lines 273-275 The authors asserted that “methylation of each gene was an essentially independent event”, this comes out of the investigations carried out on the patients in question affected by colorectal cancer. It is important to underline how, given the different context in which cells can find themselves, for example inflammatory stage, their genic and genomic methylation status can also be influenced by something else that is not strictly linked to the oncological condition, but also simply to a variety in the activity of DNMTs and TETs for example. At the same time, it is also known in the literature (PMID: 36635363; PMID: 37571432; PMID: 38612845) that lifestyle, for example food, and therefore the presence of specific molecules, or potential pollutants in water, can directly or indirectly contribute to the status of genetic and genomic methylation, collaborating with the SAM-DNMT complex; this is the reason why methylation status and also its variations can be considered multifactorial dependent. We therefore recommend to more discuss this aspect in the text, also by inserting the above-indicated references.
Hypermethylation was observed simultaneously in multiple genes in some samples and in a single gene in others, and it was assumed that methylation of each gene was an essentially independent event.
A2. The statement that "the methylation of each gene was an essentially independent event (line 294)" seems to be clear from the fact that, as shown in Tables 2 and 3, the methylation of each gene did not occur in a coordinated manner within a single sample, but rather in a distributed manner across individual samples.
As you point out, the factors that influence DNA methylation are complex and it is possible that they also influenced the results of this study. We have added 341-349 lines of text on this topic as below and included the cited paper.
“In addition, the level of methylation may be influenced by individual differences in the activity of genes involved in DNA methylation, such as DNA methyltransferase or ten-eleven translocation (TET) mediated DNA demethylation. It is also known that lifestyle, including diet, and therefore the presence of certain molecules or potential contaminants, can directly or indirectly contribute to the status of DNA methylation [27-29]. There have also been reports that the microbiota, which refers to the composition and abundance of microorganisms, affects DNA methylation in colorectal cancer[30]. As shown, the factors involved in DNA methylation are extremely diverse and it is extremely difficult to standardize them across patients.”
Q3. Line 445 It would be better to check protein expression via WB.
A3. As you pointed out, analysis at the protein level will be important in future research. we have added the following to lines 496-501 of the text.
“To understand the true impact of this complex hypermethylation on the mRNA and protein expression of tumor suppressor genes, further detailed research with larger sample sizes and optimal sample processing is needed. In addition to the NGS analysis methods demonstrated in this study, this research will require comprehensive analysis using multiple methods including RNA-seq, immunohistochemistry, and Western blotting.”
Through this revision, we feel that we could improve our article. Thank you very much for your valuable suggestions.
Sincerely,
Yo Niida, M.D., Ph.D.
Center for Clinical Genomics
Kanazawa Medical University Hospital
- Daigaku, Uchinada, Kahoku, Ishikawa, 920-0293, JAPAN
Phone No: +81 076-286-2211
Email Address: niida@kanazawa-med.ac.jp

Round 2
Reviewer 1 Report
Comments and Suggestions for Authors
Thanks to the authors for their responses to my comments in the review of their paper, which were satisfactorily addressed. I accept this publication.
Reviewer 4 Report
Comments and Suggestions for Authors
Manuscript well improved